REGISTERED REPORT PROTOCOL

# Registered report: How open do you want your science? An international investigation into knowledge and attitudes of psychology students

Hannes Jarke [1,2]*, Lea Jakob [1,3,4], Lana Bojanić [1,5], Eduardo Garcia-Garzon [1,6], Silvana Mareva [1,7], Augustin Mutak [1,8], Jovana Gjorgjiovska[1]

1 Junior Researcher Programme, Cambridge, United Kingdom, 2 Centre for Business Research, Judge Business School, University of Cambridge, United Kingdom, 3 National Institute of Mental Health, Klecany, Czech Republic, 4 3rd Faculty of Medicine, Charles University, Czech Republic, 5 Division of Psychology and Mental Health, Faculty of Biology, Medicine, and Health, University of Manchester, Manchester, United Kingdom, 6 Facultad de Salud, Universidad Camilo José Cela, Madrid, Spain, 7 MRC Cognition and Brain Sciences Unit, University of Cambridge, Cambridge, United Kingdom, 8 Methods and Evaluation/Quality Assurance, Freie Universität Berlin, Berlin, Germany

* hj318@cam.ac.uk

## Abstract

The use of Open Science practices is often proposed as a way to improve research practice, especially in psychology. Open Science can increase transparency and therefore reduce questionable research practices, making research more accessible to students, scholars, policy makers, and the public. However, little is known about how widespread Open Science practices are taught and how students are educated about these practices. In addition, it remains unknown how informing students about Open Science actually impacts their understanding and adoption of such practices. This registered report proposes the validation of a questionnaire. The aim is to survey how much psychology students know about Open Science and to assess whether knowledge of and exposure to Open Science in general—be it through university curricula or social media—influences attitudes towards the concept and intentions to implement relevant practices.

## Introduction

Open Science (OS) encompasses practices aiming to improve research through openness, transparency, rigor, reproducibility, replicability, and accumulation of knowledge (for an annotated reading list see Crüwell and colleagues [1]). The concept has received considerable recognition over the last decade: According to the Organisation for Economic Co-operation and Development [2], OS could improve efficiency in science, increase research transparency and quality, accelerate knowledge transfer, address global challenges more effectively, and promote the public's engagement with science and research. Encouragement to implement OS practices has also come from the European Commission [3] in 2018 in the form of the *Open*

**Data Availability Statement:** All pilot data is available via https://osf.io/n89xs/.

**Funding:** The author(s) received no specific funding for this work.

**Competing interests:** The authors have declared that no competing interests exist.

*Science Agenda*, which outlines a set of recommendations towards wider OS implementation. In 2020 we have further seen a rise of the use of OS practices across disciplines aimed to enable fast and efficient sharing of findings and collaboration regarding the COVID-19 pandemic [4, 5].

The FOSTER model and taxonomy [6] describes OS on six dimensions. *Open Access* refers to online, free of cost access to peer-reviewed scientific content with limited copyright and licensing restrictions. *Open Data* comprises online, free of cost, accessible data that can be used, reused, and distributed, provided that the data source is attributed. *Open Reproducible Research* refers to the act of practicing OS within one's research and providing free access to experimental elements for research reproduction. *Open Science Evaluation* is a form of research results assessment which is not limited to few anonymous peer-reviewers but requires the community's contribution. *Open Science Policies* describe best practice guidelines for applying OS and achieving its fundamental goals. Finally, *Open Science Tools* can assist in the process of delivering and building on OS, such as Open Online repositories.

Some have expressed criticism of OS or specific dimensions of the concept, ranging from the rise of predatory journals, the time and cost it takes to follow the practices of Open Research, or the perception of "data parasites" who unequally benefit from using Open Data provided by others [7]. However, other authors point out the benefit of thinking about the analysis before approaching data collection [8] and using it as an educational tool [9–12].

In psychology, the OS movement and the practices it promotes have been brought to the center of attention in light of the replication crisis (i.e., many well-known studies do not replicate) that the field is facing [13]. The replication crisis has brought a psychology renaissance mostly reflecting the growing adoption of OS principles (see [14] for a summary of events and developments). Institutional efforts to support OS implementation—such as those by the Center for Open Science (see https://www.cos.io/about/mission)—are growing in number, and preliminary evidence suggests that they are successful in attracting researchers' attention [15]. Recent OS outputs include materials with both practical and didactic value such as the Transparency Checklist [16] that point authors towards the most important steps needed for transparent research.

Investment in the education of junior researchers in psychology is a prerequisite to ensure the successful implementation of such practices. In a paper on the future of the OS movement, Spellman [17] points out how valuable web resources are to achieve this goal. Currently, there are many online tools dedicated to educating and encouraging young researchers to adopt OS practices (e.g., Open Science Framework, Coursera). However, while there is some evidence of OS principles being adopted more and more [e.g., 18, 19], it is unknown how far they are incorporated into the formal education of students of psychology and behavioral sciences. Spellman, Gilbert, and Corker [20] speculate that: "young researchers have been trained in a larger and more diverse set of labs that, we suspect, vary more in procedures and assumptions than those from the past" (p.15). Funder et al. [21] recommend encouraging a culture of "getting it right" via undergraduate courses, textbooks, and workshops while stressing the importance of modelling best practices by example to junior researchers. In practice, however, it seems that the advancement in training is currently limited and predominantly informal, mostly pertaining to one-on-one training between early-career researchers and their supervisors or mentors [13]. The European Commission [22] has recognized this situation and highlights the comprehensive training of researchers in OS practices as one of its key recommendations to increase the use of OS.

Given the growing acknowledgement that efforts should be concentrated towards the education of junior researchers, with findings that preregistration is perceived positively by students as a helpful tool for planning research [9], we argue that learning and adopting OS

practices has the potential to benefit all psychology students—regardless of their future career aspirations—in their understanding of how empirical evidence is gathered, analyzed, and interpreted. Even if graduates will be applying research findings in practice instead of producing them, they should be able to recognize research strengths and weaknesses and may benefit from accessing data and materials for self-learning throughout their careers. Consistent with this position, evidence shows that the adoption of open educational resources, especially in the form of electronic textbooks, can have beneficial effects on learning outcomes during students' education [23].

Across countries, universities hosting psychological studies are likely to differ in their educational practices. In a report on public funding, the European University Association finds a widening divide in universities' funding across European countries [24]. Universities struggling with budget and staff reductions may be slower in the adoption of some aspects of OS [25], even though using free materials may be beneficial especially when working under strong budgetary constraints. Language could be another potential barrier to the spreading of the movement, as the majority of OS resources currently appear to exist primarily in English.

The aims of this investigation are twofold: Firstly, we aim to investigate to what extent psychology students are exposed to OS practices, both in their curricula, as well as online. Our focus here is on gaining insight into how widely OS practices are known and taught. Secondly, we want to explore how exposure to OS practices impacts students' attitudes and intention to implement them in practice. We created a questionnaire which we aim to validate through this investigation. The current results could provide a preliminary estimate about how widespread the teaching of OS practices is at university level and if it is related to intention to adopt OS practices. Future uses of the questionnaire could enable more focused inquiry into specific training programmes and populations (graduate vs. undergraduate etc.).

## Research questions

**RQ1.** To assess the validity and factor structure of a newly developed questionnaire exploring OS knowledge and attitudes among psychology students.

**RQ2.** To understand the extent to which psychology students are exposed to OS practices within, as well as outside their formal educational curricula.

**RQ3.** To assess how exposure to OS practices within and outside formal education impacts students' attitudes towards them, including the intention to implement them.

**RQ4.** To gain insight into the role that OS-related online content consumption (i.e., social media) and interest in research play in improving attitudes towards OS activities and their general knowledge.

**Exploratory.** To explore whether attitudes towards OS and knowledge rates differ across universities and/or countries.

## Hypotheses

**Hypothesis 1.** Research question one is exploratory in nature. However, despite a lack of data, it is our expectation that—given that OS practices are rather new and only some students are focusing on a career in research—most students would be relatively unfamiliar with them. Either way, it is expected that as students advance in their studies, they could become more acquainted with them in their formal or informal education, and present higher rates of OS knowledge.

**Hypothesis 2.** We expect exposure to OS to be highly dependent upon university and country engagement on OS practices. Accordingly, institutions that include OS in their

curriculum would present larger shares of students with some knowledge of these practices and more positive attitudes towards them.

**Hypothesis 3 & 4.** We expect a positive, directional relation regarding the relationship between attitudes towards OS and OS knowledge. As remarked by previous research, students exposed to OS activities tend to have a more favorable view of OS practices. Additionally, students familiar with ongoing scientific controversies (e.g., replication crisis) would also tend to have more positive attitudes and knowledge of OS practices.

**Hypothesis 5, 6, & 7.** We expect that research-related social media consumption and research interest will have a positive effect on attitudes towards OS and OS knowledge. Particularly, students with higher engagement rates with social-media content related to OS would present more knowledge and more positive attitudes towards OS activities. Additionally, students with a higher interest in research, or with previous research experience are expected to have more positive OS attitudes and higher knowledge as well. Lastly, students interested in research are also expected to be more likely to use social media to be informed about this topic. Thus, we expect students interested in research to have higher consumption rates of research-related social media, and more positive OS attitudes and knowledge.

## Data, materials, and online resources

All data, analyses, code, and results will be published in an accompanying online repository (https://osf.io/n89xs/). The questionnaire was part of the supporting information of this pre-registration made available to reviewers. It will be added to publicly available resources upon publication of the final paper.

### Reporting

We report how we determined our sample size, all data exclusions, all manipulations, and all measures in the study (see also [26]) after presenting pilot data.

### Measure

We designed a new questionnaire for measuring OS knowledge and adoption across psychology students for this study (this will be made publicly available after data collection in the accompanying OSF repository [https://osf.io/njqvg/] and in the final paper). This measure was created to gain insight into the OS knowledge and attitudes towards OS across psychology students, as well as interest in research and OS-related social media usage. With the exception of "Part IV: Attitudes towards OS", where we include some questions developed by Orion Open Science [27], items were specifically developed for this study. Nevertheless, many items present in our scales could have a large overlap with existing questionnaires in the field (i.e., social media usage scales).

To ensure consistency across all measurement blocks, all Likert-scale items were designed to follow a positively worded, unipolar structure in a 1-to-5 scale (where one represents a lack of endorsement of the measured construct and five, strong agreement) with adapted verbal anchors. Two attention check items were introduced in Part IV and Part V of the questionnaire. In these items, participants were instructed to mark the middle point of a 1-to-5 Likert scale with similar verbal anchors than the previous and the following items. In detail, the complete measure constitutes five main blocks:

**Sociodemographic background.** Demographic data collected will include age and gender. Additionally, participants' main university (alma mater), the degree in which the participant is currently enrolled [Bachelor / Master / PhD / Other], year of study at the university level, and participation in exchange programs [Yes/No and if Yes, hosting university], and self-rated

level of English is recorded. No further personal data (i.e., personal email) will be collected, preventing individual participant identification.

**Part I. Social media usage.**   Participants will be asked to provide an estimate of how many hours per day they engage in social media use. They will further be questioned about their engagement with research-related social media content: Whether they follow psychological science/scientists on social media; whether they listen to research-related podcasts/follow blogs/participate in online discussion groups; and whether they are currently registered on research-oriented platforms such as ResearchGate or Academia.edu.

**Part II. Interest in psychological research.**   Participants will be asked with regards to: a) general interest in psychological research; b) current interest in engaging with research-related activities; c) previous or current participation in research projects and research area; d) interest in pursuing an academic career; e) engagement with scientific literature in their leisure.

**Part III. Attitudes towards OS.**   The block of attitudes towards OS is adapted from the *Analysis and Benchmarking*: *Self-Assessment Questionnaire* from the Orion Project [27] blocks 2, 3, 5, 6, as well as a final question measuring participants' overall self-described stance on OS. However, the response scale was changed to be consistent with previous questions in our newly developed measure (1-to-5 Likert response) and questions were adapted to the student population. Additionally, the main question in block 5 was changed from "*In your opinion, why should science be open*?" to "*In your opinion, which of the following areas could be improved through a more open and accessible science*?" to improve item clarity in the target population.

**Part IV. Open Science knowledge.**   This block of items contains questions asking for previous involvement with OS practices in the form of practical experience or instructions/ courses, participants' desire for OS elements to be included in their academic curriculum, and how knowledgeable they consider themselves with regards to OS. Participants are then asked to select which OS dimensions were taught in their courses and to rate their knowledge regarding these topics. Some of these OS activities (i.e., Open data, access to publication and education) were also adapted from the *Analysis and Benchmarking*: *Self-Assessment Questionnaire* [27]. Further questions aim at assessing the state of OS at their current institution, an estimate of how easy it would be for students to implement OS in future projects such as their thesis, and whether they would consider doing so. The block closes with a question regarding their awareness of the ongoing replication crisis.

## Data exclusion

Participants not completing the survey, presenting aberrant response patterns (i.e., all responses being 1 or 5) or response times of over one hour (which equals more than twice of the assumed average response time) or less than five minutes will be removed from the sample. Participants who do not pass the attention check or do not answer the first item with a valid university name will be removed as well.

## Methods

### Participants

The target population encompasses English speaking psychology students currently enrolled in either Bachelor or Master studies (or comparable, e.g., Diploma), as well as doctoral students, independent of country of study. Students from both private and public universities would be considered as participants for this study.

We will collect a convenience, non-weighted sample of English-speaking students using the authors' personal and academic networks, as well as recruiting via social media, including Facebook, Twitter, Reddit, and email lists. There will be no compensation for participation in

the study. Participants have to declare being enrolled in a university-level psychology or behavioral science degree.

## Ethical approval

Data collection for the pilot sample was, and main data collection will be, carried out in accordance with the *Declaration of Helsinki*. No negative impact is expected on participants that exceed typical daily emotional experiences. The questionnaire has been kept as short and concise as possible to limit the necessary time investment of participants to the minimum. This study received ethical approval from the Cambridge Psychology Research Ethics Committee at the Council of the School of the Biological Sciences, University of Cambridge before pilot investigation commenced.

## Procedure

Data collection will be performed using Qualtrics to collect and store data. Participants will be informed that this study investigates opinions and knowledge about research practices. They will further be notified about roughly what kind of data we will collect, that they are not supposed to provide any personal identifiable data, and will be provided with contact details for the primary investigator of the study. It will also be specified that participants are under no obligation to finish the questionnaire once started and may abort the process whenever they wish. Once the participant accepts the conditions expressed in the informed consent, they will be presented with the questionnaire. We initially expected that the completion of the questionnaire would take no longer than 20–25 minutes. Our pilot data support this assumption, with the Mean duration being 26 Minutes and 22 Seconds, and the Median 21 Minutes and 51 seconds. After completing the questionnaire, participants will be provided with links providing further information on OS.

## Analysis plan

### Descriptive statistics

We will calculate descriptive statistics on the demographic data to gain insights into the characteristics of our sample. To gain answers related to H1, we will calculate descriptive statistics on variables indicating OS knowledge. Our analyses will be performed on original variables, but also on derived variables.

**Original variables.** For categorical variables, we will compute the percentage of participants belonging to each category. Depending on the size of groups within the sample, these would ideally be stratified by university affiliation.

For continuous variables, we will first inspect if data is normally distributed using the Anderson-Darling test. If the distributions are normal, we will compute the means and standard deviations of the corresponding variables; if they are not, we will compute their medians and interquartile ranges.

**Derived variables.** We will derive some variables based on the original variables in our data. All derived variables will be continuous and therefore, the same process that was described above will be applied. We will derive the following variables through calculating sum scores from the respective set of questions:

- Accessibility to scientific community

- Accessibility to the public

- Areas that can be improved by OS

- Areas that cannot be improved by OS

- OS Knowledge

- General attitude towards OS

## Inferential statistics

We will use inferential statistics to answer our research questions related to the connection between exposure to OS and prevalence of OS practices/attitudes. A first question to be addressed is to understand data structure: as individual responses are nested within their class, year of study, university, or country. To account for such dependencies, random intercept multilevel regression will be used to decompose variance into each level component (using the intraclass correlation coefficients). We only plan to explore levels where a sufficient number of individuals are available for ensuring adequate variance estimation (e.g., at university level, at least 20–25 students are available). If results suggest that such dependencies exist at any level, we will take them into account in our analyses.

To gain answers related to H1, we will calculate descriptive statistics on variables indicating OS knowledge. We will also correlate OS knowledge with the number of years studying. A significant positive correlation will be taken as a confirmation of the hypothesis.

To test H2, we will perform a series of t-tests to discern whether various aspects of OS knowledge and attitudes are dependent on the inclusion of OS in the university curriculum. In line with the hypothesis, we expect all t-tests to be significant and we expect that universities with OS in their curriculum will have higher scores than universities with no OS in their curriculum.

H3 will be tested by calculating correlations between various aspects of OS attitudes and OS knowledge. We expect correlations to be significant and positive to confirm the hypothesis.

For H4, we will use a series of t-tests to explore if students who have indicated that they have heard about the replication crisis and who did not select incorrect statements related to the replication crisis differ in various aspects of OS attitudes and knowledge from students familiar with the replication crisis.

For H5 and H6 we will perform two series of linear regression analyses. In the first series, social media consumption, research interest and previous research experience will be predictors, whereas different aspects of OS knowledge and attitudes will be criterion variables in different models.

To test H7, we will correlate research-related social media activity with interest in research and with research experience.

We will conduct a number of preliminary analyses that will determine if some of the further analyses we have planned are feasible. We will start by conducting exploratory factor analyses (EFA) on items from Parts II, III, and V to discern whether it is possible to calculate scale scores that would indicate higher usage of research related social media, higher interest in research, and higher knowledge of OS, respectively. More specifically, we will conduct a single-factor EFA on each of the parts representing constructs above.

We will consider scale scores derived variables. Should a scale happen to contain only items that are continuous variables, we will compute the scale score as a simple sum of item scores. We will inspect the quality of these solutions in terms of factor's loadings and model fit, using omega categorical [28] to assess their reliability. For an unidimensional model, it estimates the reliability of observed total sum scores more adequately than alternatives such as Cronbach's alpha while taking into account the categorical nature of the items used. If unidimensional models are not suitable, we will explore multidimensional alternatives including a general

factor, using omega hierarchical for the estimation of the total observed scores in these scales [29]. In both cases, values over .80 are considered as adequate.

However, if a scale contains a mix of continuous and categorical items, we will calculate Bartlett's factor scores [30] instead. In these cases, we will assess the quality of the derived scores evaluating factor determinacy. Values close to .90 indicate that factor scores are sufficiently determined for their posterior use [31].

Furthermore, we will compute a Structural Equation Model (SEM) in which we assume that all the latent variables (usage of social media, interest in research, attitudes towards OS and knowledge of OS) are correlated. Modifications to the model will be made if the procedure described in the previous paragraph shows that some scale scores cannot be computed (those latent variables will be excluded from the model). The model will also include demographic and other variables relevant to the hypotheses (years studying, inclusion of OS in the curriculum, knowledge of replication crisis).

## Pilot study data and insights

We conducted a pilot study on a small convenience sample ($N = 38$, $n = 30$ after removing incomplete and nonsensical submissions) to determine whether the questionnaire was easily understandable and to gain initial insights into feasibility. In response to observed feedback, the following changes were introduced with regards to the questionnaire and analysis plan:

- Responses to all items are now required by the survey system ("force response") to ensure no items are overlooked.

- Item clarity has been improved in several ways: For example, negative items (e.g., "In your opinion, which of the following areas could **not** be improved through a more open and accessible science?") present the part "not" bolded; (b) all double-negatives have been removed.

- Failing to provide an existing institution of study or work has been made an exclusion criterion—we assume that if a participant fails to provide a valid answer to the very first item, they will not provide reliable data.

- We found that asking for a field of specialty as an open-ended question generated too many different answers. As such, we will limit answers to generalized areas in a multiple-choice format from a comprehensive list.

We are highlighting some preliminary insights gained from the pilot investigation, but make the full dataset, as well as all analyses available in the accompanying repository. Participant feedback did not include any issues with understanding language or phrasing despite more than 75% of participants being non-native speakers, which supports our assumption of the survey being easily understandable to the target population. Within the sample, all OS dimensions are reported to be taught in curricula by 50% or less of participants (Fig 1). While this does not warrant any inferences, a similar distribution in the main sample would constitute near-ideal conditions for comparison.

Contrary to our key hypothesis, we find no evidence that exposure to OS in university courses impact OS knowledge (Fig 2) or attitudes towards OS (Fig 3).

We find 45 statistically significant positive correlations between single items used to measure self-described OS knowledge (see Fig 4).

We conducted multiple EFA's to gain preliminary insights about the questionnaire. The commented code and results are available at the accompanying online repository. We find evidence pointing toward a singular factor for *Interest in Research*, the existence of three factors

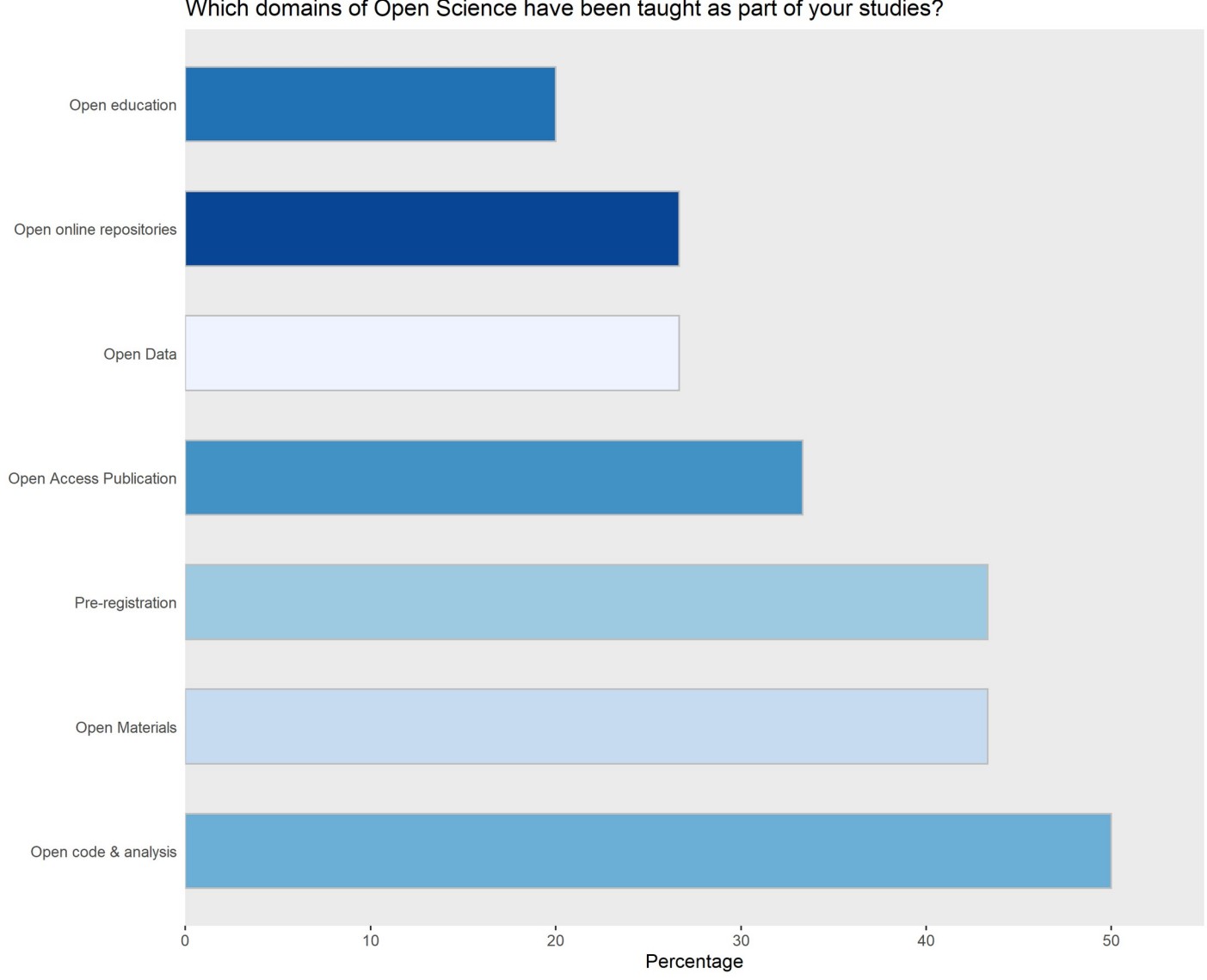

**Fig 1. The relative number of OS dimensions taught in university curricula.**

within the section *Open Science Knowledge* (general OS knowledge, self-assessment of OS knowledge, Open Materials and Code), but no evidence for a suitable factor model for *Social Media Usage*. It is important to note that this analysis was mainly conducted as an example underlying this registered report. Results should not be taken at face-value, as measures for sampling adequacy and sphericity suggest that the pilot data is not quite suitable for factor analysis.

## Sample size estimation

We studied sample size requirements to observe sufficient power in all our pre-planned analyses: a) bilateral Welch's t-test and correlation analyses between observed variables; b) latent variable correlations in a SEM model. We will set our target sample size to the minimum sample size required to observe a .80 power in either analysis.

## Academic Teaching and OS Knowledge

$(27) = 1.04$, $p = 0.310$, $= 0.20$, $CI_{95\%}$ [-0.18, 0.52], $n_{pairs} = 29$

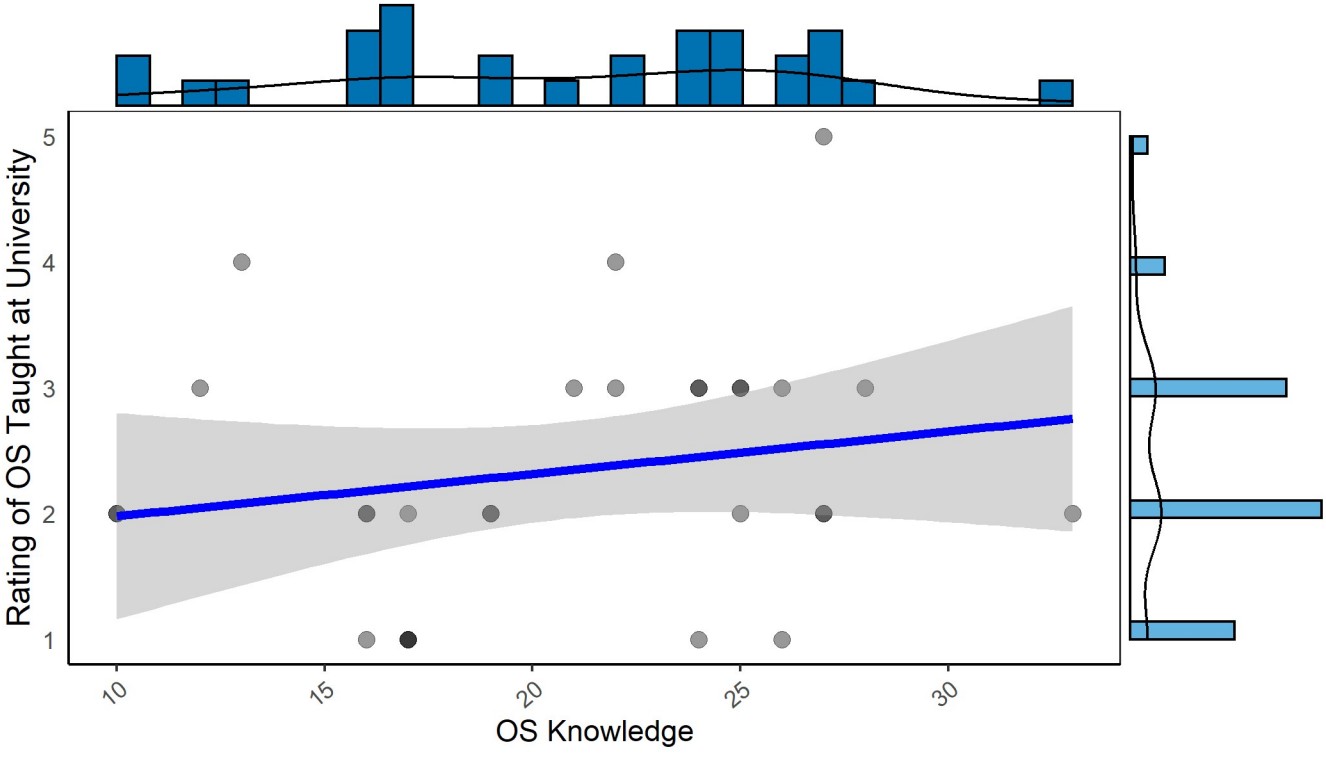

$\log_e(BF_{01}) = 0.79$, $\hat{\rho}_{median}^{posterior} = 0.17$, $CI_{95\%}^{HDI}$ [-0.17, 0.50], $r_{Cauchy}^{JZS} = 1.41$

**Fig 2. Correlation of OS knowledge and OS being taught at university.**

With regards to Welch's t-test and observed correlations, we conducted two different simulations. We simulated 500 datasets from both normally distributed and Likert scales (from a five-point discretized normal latent distribution) in each combination of sample size (from 50 to 1000 at 50 participants jumps) and standardized effect size (from .05 to .80 at jumps of .05 units). As pilot data results suggested that effect sizes (Cohen's $d$ and $r$) as small as .20 could be observed, we were interested in establishing the necessary number of participants to achieve a power of .80 under those conditions. Results indicated that sample size requirements were higher for Likert-scales than for the continuously distributed variables and higher for Welch's t-test than for observed correlations. Thus, we estimated our target sample size for the most conservative cases (Welch's t-test conducted using Likert-scale data). Results suggested that at least 300 participants would be necessary to achieve .80 power (Fig 5).

For the SEM case, we largely followed Wang & Rhemtulla's approach to estimate power analyses in SEMs [32]. We conducted a small simulation study varying three main conditions: a) sample size varied from 100 to 1000 participants at 50 participants jumps; b) factor loadings were simulated from a uniform distribution in a low factor loading (.30 to .50) or high factor loading condition (.50 to .70) conditions; c) average latent correlation size, including a low correlation ($r = .30$) and a high condition ($r = .50$).

## Academic Teaching and OS Attitude

(26) = 0.37, $p$ = 0.717,  = 0.07, $CI_{95\%}$ [-0.31, 0.43], $n_{pairs}$ = 28

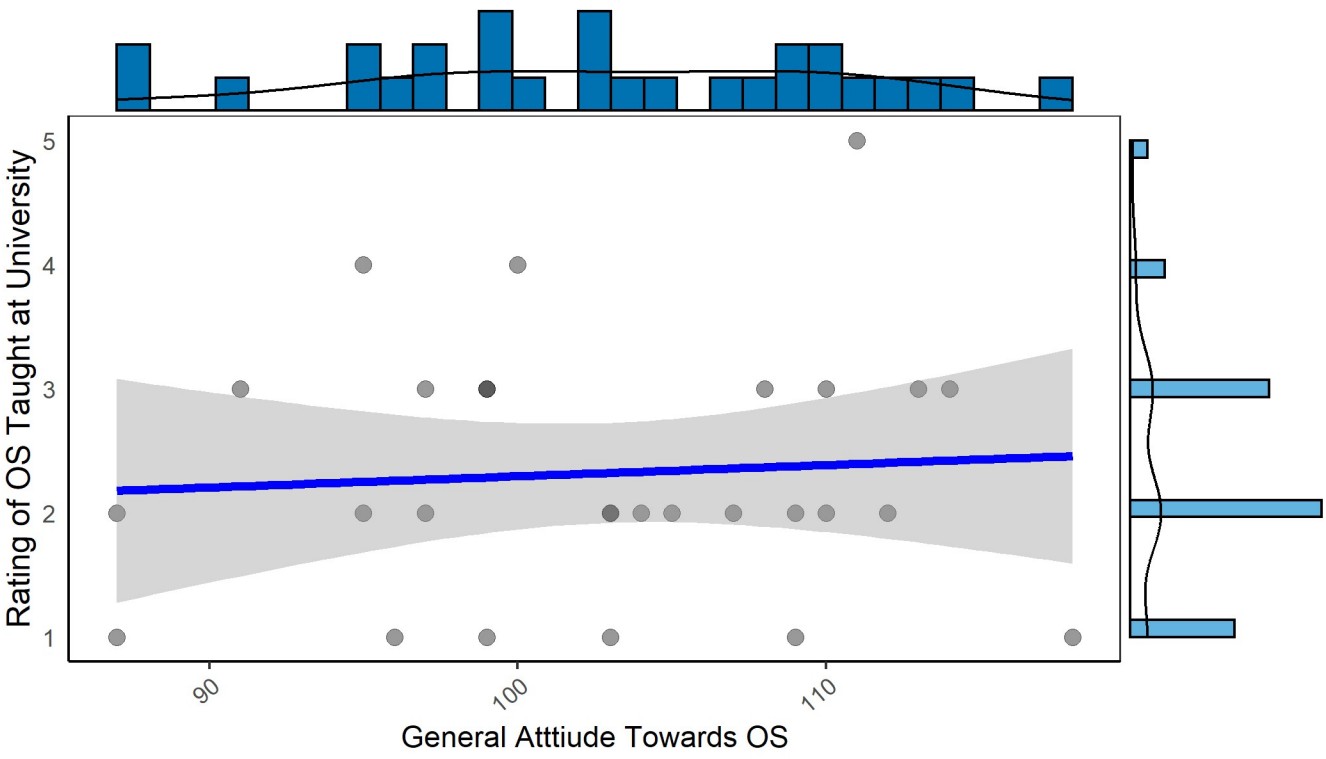

$$\log_e(BF_{01}) = 1.20, \hat{\rho}_{median}^{posterior} = 0.06, CI_{95\%}^{HDI} [-0.29, 0.40], r_{Cauchy}^{JZS} = 1.41$$

**Fig 3. Correlation of OS being taught at university to participants' attitude towards OS.** A positive correlation displays more positive attitudes towards OS when present as part of the curriculum.

The estimated SEMs were defined to resemble our model of interest, including six latent correlations among the main factors: use of social media (four items), research interest (four items), a general attitude towards OS factor, and a general OS knowledge factor. The general attitude towards OS was defined using four first-order factors: attitudes against OS (5 items), attitudes towards participation in OS (5 items), openness in areas (8 items), and negative attitudes towards OS (7 items). The general knowledge factor is measured by two first-order factors: OS dimension knowledge (7 items) and OS knowledge rating (7 items). All items were estimated as continuous variables. We estimated 300 sample matrices of each condition and estimated them using maximum likelihood. Estimated power was averaged within condition to reflect the correct estimation of the six possible latent correlations. For example, if for a given condition and replication, only five out of six correlations were significant, power was estimated as .83 for that replication.

Three main results were observed from the simulation (Fig 6): a) average latent correlation size was more important for power estimation than average factor loading size; b) under high average factor loading and high latent correlations, even sample sizes as small as 200 resulted in power estimation over .90; c) On the other hand, when both, average factor loading and latent correlation, were low, sample sizes over 1000 were necessary to achieve power over .80.

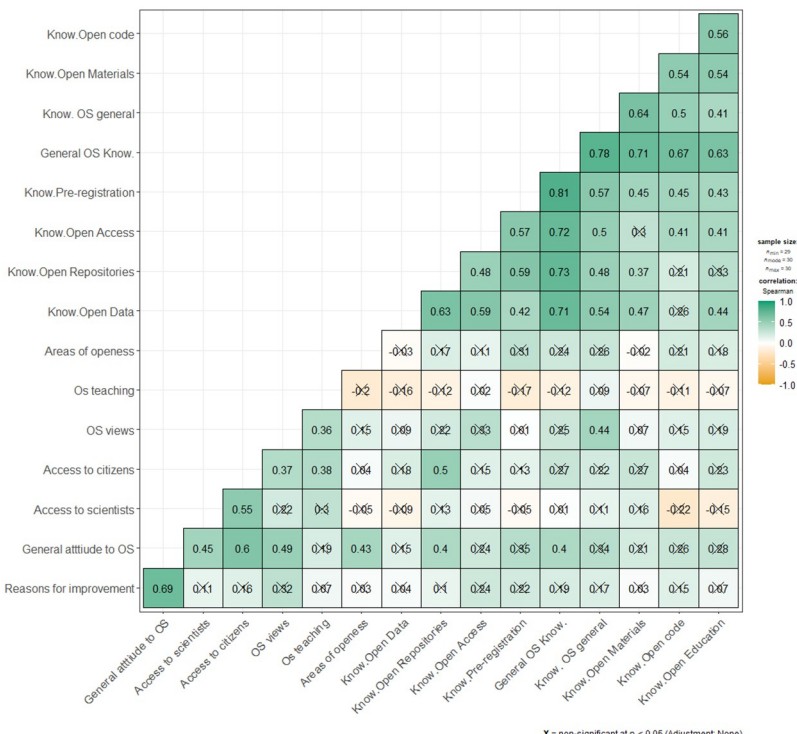

**Fig 4. Correlations between items used to measure OS attitudes and knowledge.**

Nevertheless, when sampling around 400 individuals, power over .80 was achieved even if factor loadings were high. This held true even for low true latent correlations.

To summarize, we obtained evidence from different sample size estimations that we should aim to obtain a minimum sample size of at least 400 participants to detect all effects reliably.

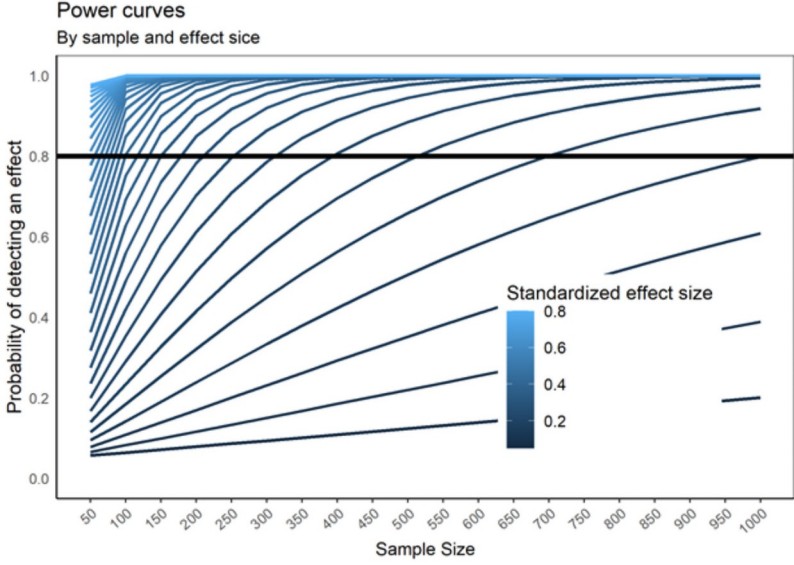

**Fig 5. Sample size needed to detect effects of d = 0.2 and higher.**

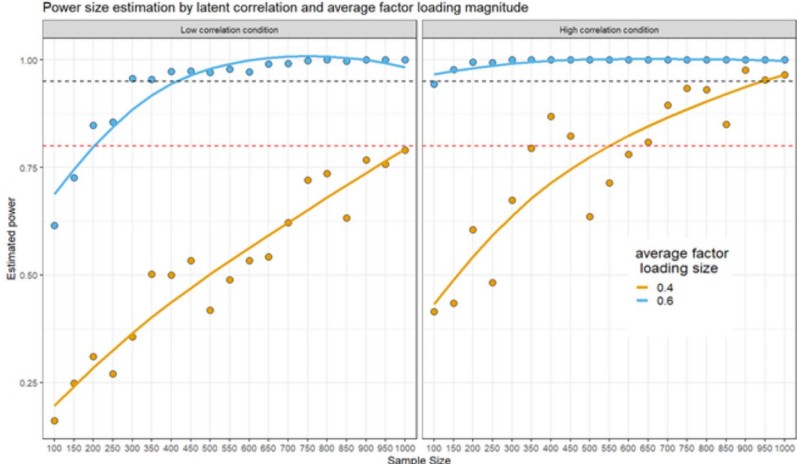

**Fig 6. Simulations for power analysis for the SEM model including six latent correlated factors.**

We will collect data for a set time of three months. Should data of at least 440 participants have been collected by then (adding 10% to 400, accounting for the event that some participants may have to be excluded), data collection will stop. If this goal has not been reached by then, we will continue data collection until a sample of at least 440 participants has been reached and stop once the minimum sample size is met.

## Limitations and anticipated risks

We intend to employ convenience sampling. As such, representativeness for each university is questionable. Those students who have already heard about Open Science and who have more positive attitudes towards it may also be more inclined to participate in the study than others. Similarly, as our recruitment strategy is focused on English-speaking students, this might result in a potential barrier to the participation of students whose native language and/or language of study is not English. While all of this limits the generalizability of results, we would like to highlight that this study is conceptualized as the first step aimed to investigate the validity and reliability of the measure. Should results be encouraging, we will lead efforts to translate the questionnaire and apply it to multiple countries and more diverse populations of psychology students. Finally, the data we collect relies mainly on self-reports.

## Future plans

This investigation is conceptualized to be the first step in assessing the research questions more widely and internationally, with the goal of future studies including translated versions of the questionnaire being distributed to representative country samples, allowing comparisons between institutional curricula featuring various levels of OS teaching and exposure. More than 50 collaborators have already signaled interest in adopting the questionnaire across 23 countries. This study serves to validate the questionnaire first—and potentially adopt, based on results—before a widespread, multi-country effort may be undertaken.

## Acknowledgments

We would like to acknowledge Professor Christopher Chambers for his input at the early stages of this project, specifically related to his advice about creating comprehensive registered

reports. We would also like to thank Dr Gianmarco Altoè, Dr Ughetta Moscardino, Giulia Bertoldo, and the *Psicostat* group at the University of Padova, who organized an insightful Winter School in 2019, without which this project would not exist. Lastly, we are grateful to the *Junior Researcher Programme* for bringing us together and inspiring us to continue these kinds of international and progressive investigations.

## Author Contributions

**Conceptualization:** Hannes Jarke, Lea Jakob, Silvana Mareva.

**Data curation:** Hannes Jarke, Lana Bojanić, Eduardo Garcia-Garzon, Augustin Mutak, Jovana Gjorgjiovska.

**Formal analysis:** Lana Bojanić, Eduardo Garcia-Garzon, Augustin Mutak.

**Investigation:** Hannes Jarke, Lea Jakob, Lana Bojanić, Eduardo Garcia-Garzon.

**Methodology:** Hannes Jarke, Lea Jakob, Lana Bojanić, Eduardo Garcia-Garzon, Silvana Mareva, Augustin Mutak.

**Project administration:** Hannes Jarke, Silvana Mareva.

**Resources:** Hannes Jarke, Jovana Gjorgjiovska.

**Supervision:** Hannes Jarke.

**Visualization:** Eduardo Garcia-Garzon.

**Writing – original draft:** Hannes Jarke, Lea Jakob, Lana Bojanić, Eduardo Garcia-Garzon, Silvana Mareva, Augustin Mutak, Jovana Gjorgjiovska.

**Writing – review & editing:** Hannes Jarke, Lea Jakob, Lana Bojanić, Eduardo Garcia-Garzon, Silvana Mareva, Augustin Mutak, Jovana Gjorgjiovska.

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
