## [Decision Letter · Decision Letter 0]

29 Jul 2021

PONE-D-21-21771

How Open Do You Want Your Science? An International Investigation into Knowledge and Attitudes of Psychology Students

PLOS ONE

Dear Dr. Jarke,

Please do the minor corrections based on the reviewers' comments.

Thank you,

Or Kan Soh

Academic Editor

PLOS ONE

Reviewers' comments:

Reviewer's Responses to Questions

**Comments to the Author**

1. Does the manuscript provide a valid rationale for the proposed study, with clearly identified and justified research questions?

Reviewer #1: Yes

Reviewer #2: Yes

2. Is the protocol technically sound and planned in a manner that will lead to a meaningful outcome and allow testing the stated hypotheses?

Reviewer #1: Yes

Reviewer #2: Partly

3. Is the methodology feasible and described in sufficient detail to allow the work to be replicable?

Reviewer #1: Yes

Reviewer #2: Yes

4. Have the authors described where all data underlying the findings will be made available when the study is complete?

Reviewer #1: Yes

Reviewer #2: Yes

5. Is the manuscript presented in an intelligible fashion and written in standard English?

Reviewer #1: Yes

Reviewer #2: Yes

6. Review Comments to the Author

You may also provide optional suggestions and comments to authors that they might find helpful in planning their study.

Reviewer #1: Overall, the protocol is well planned and written. There are only a couple of small issues I would like to point out: the colour scale in Figure 5 makes it difficult to distinguish differences, and the last sentence on page 14 is missing a closing parenthesis.

Reviewer #2: the study is well designed and prepared, but the main issue is the English Language that is used especially the tenses; major parts of the study talks about the future and the study is already written and has been sent for publication. I think it is better to use past tense and present perfect as required in each place, or to give the study a language expert to review it.

I also recommend adding some idea in the topic about developing a tool for assessing attitudes about open science.

7. PLOS authors have the option to publish the peer review history of their article (what does this mean?). If published, this will include your full peer review and any attached files.

Reviewer #1: **Yes: **Justine Vandendorpe

Reviewer #2: **Yes: **Nazdar Qudrat Abas

---

## [Author Response · Author response to Decision Letter 0]

31 Jul 2021

Replies to Reviewer #1: 

R1: Overall, the protocol is well planned and written. There are only a couple of small issues I would like to point out: the colour scale in Figure 5 makes it difficult to distinguish differences, and the last sentence on page 14 is missing a closing parenthesis.

Reply: Thank you very much for your feedback, we are glad to see that our protocol is received as well structured, and the content is clear to people outside the team. 

We have fixed the issue with the parenthesis on page 14.

We would like to thank you for your comment on Figure 5. We would like to clarify that the gradient colour is only an addition, which we hoped would make it more intuitive to read, but not technically necessary. As the differences in colour strength are determined by the data, there is, unfortunately, nothing we can do to distinguish the lines more without misrepresenting the data. Nevertheless, we hope that the graphic, together with the in-text explanation sufficiently explain this aspect of the sample size estimation.

Replies to Reviewer #2:

R2: the study is well designed and prepared, but the main issue is the English Language that is used especially the tenses; major parts of the study talks about the future and the study is already written and has been sent for publication. I think it is better to use past tense and present perfect as required in each place, or to give the study a language expert to review it.

Reply: Thank you very much for your feedback and review, we are glad to learn that you find our study design in the registered report protocol well designed and prepared.

As for the tenses used throughout the manuscript, we have re-checked the writing with native speakers an believe they are used correctly. As this is a protocol, most sections use future tense(s), as they describe the study plan to be carried out post-protocol publication. Only the sections on sample size estimation and the section on insights from the pilot study are written in past tense, as they refer to analyses and simulations already conducted in preparation for the main study.

R2: I also recommend adding some idea in the topic about developing a tool for assessing attitudes about open science.

Reply: The development of the tool for assessing attitudes is first described in lines 114-116 and then quantified through research questions 1, 3, and 4. We understand that there are certainly more aspects regarding the attitudes towards Open Science which would be interesting to study, but since our questionnaire is already rather long, we would prefer not to add any further items. This would also require a recalculation of all aspects with regards to power and we would have to re-run the pilot study. As such, we hope you can understand that we would like to avoid such big changes in the research design at this point.

---

## [Decision Letter · Decision Letter 1]

29 Nov 2021

How Open Do You Want Your Science? An International Investigation into Knowledge and Attitudes of Psychology Students

PONE-D-21-21771R1

Dear Dr. Jarke,

We’re pleased to inform you that your manuscript has been judged scientifically suitable for publication and will be formally accepted for publication once it meets all outstanding technical requirements.

Kind regards,

Prabhat Mittal, Ph.D.

Academic Editor

PLOS ONE

Reviewers' comments:

Reviewer's Responses to Questions

**Comments to the Author**

1. Does the manuscript provide a valid rationale for the proposed study, with clearly identified and justified research questions?

Reviewer #1: Yes

Reviewer #2: Yes

2. Is the protocol technically sound and planned in a manner that will lead to a meaningful outcome and allow testing the stated hypotheses?

Reviewer #1: Yes

Reviewer #2: Yes

3. Is the methodology feasible and described in sufficient detail to allow the work to be replicable?

Reviewer #1: Yes

Reviewer #2: Yes

4. Have the authors described where all data underlying the findings will be made available when the study is complete?

Reviewer #1: Yes

Reviewer #2: Yes

5. Is the manuscript presented in an intelligible fashion and written in standard English?

Reviewer #1: Yes

Reviewer #2: Yes

6. Review Comments to the Author

You may also provide optional suggestions and comments to authors that they might find helpful in planning their study.

Reviewer #1: I am satisfied with the answer the authors have provided in response to my initial review. In my opinion, the manuscript can be published as it is.

Reviewer #2: The open science is really necessary to be studied, this will bring better understanding for students and faculties to be able get the information they need for their studies and also to stay always up to date with the latest updates in their profession. I think you have done a wonderful job, wish you success and the best of luck.

7. PLOS authors have the option to publish the peer review history of their article (what does this mean?). If published, this will include your full peer review and any attached files.

Reviewer #1: **Yes: **Justine Vandendorpe

Reviewer #2: **Yes: **Nazdar Quudrat Abas

---

## [Editor Report · Acceptance letter]

6 Dec 2021

PONE-D-21-21771R1 

Registered report: How open do you want your science? An international investigation into knowledge and attitudes of psychology students 

Dear Dr. Jarke:

I'm pleased to inform you that your manuscript has been deemed suitable for publication in PLOS ONE. Congratulations! Your manuscript is now with our production department. 

Kind regards, 

on behalf of

Dr. Prabhat Mittal 

Academic Editor

PLOS ONE